Genetic signatures of population bottlenecks, relatedness, and inbreeding highlight recent and novel conservation concerns in the Egyptian vulture

Blanco Guillermo gblanco@mncn.csic.es g.blanco@csic.es
Morinha Francisco
Department of Evolutionary Ecology, National Museum of Natural Sciences (MNCN), Spanish National Research Council (CSIC) , Madrid , Spain
Wink Michael
Electronic publication date: 2021 Mar 25
Publication date: 2021
Volume: 9
Electronic Location ID: e11139
Received 2020 Dec 4; Accepted 2021 Mar 1
Copyright: ©2021 Blanco and Morinha
Copyright year: 2021
Copyright holder: Blanco and Morinha
License: This is an open access article distributed under the terms of the Creative Commons Attribution License, which permits unrestricted use, distribution, reproduction and adaptation in any medium and for any purpose provided that it is properly attributed. For attribution, the original author(s), title, publication source (PeerJ) and either DOI or URL of the article must be cited.
License URL: https://creativecommons.org/licenses/by/4.0/

Keywords: Genetic diversity, Bottlenecks, Inbreeding, Feather malformations, Neophron percnopterus

Funding: Spanish Ministry of Science and Innovation CGL2009-12753-C02-01/BOS CGL2010-15726 PID2019-109685GB-I00 Spain’s Ministry of Science and Innovation FJCI-2017-32055 Funds were provided by the projects CGL2009-12753-C02-01/BOS, CGL2010-15726 and PID2019-109685GB-I00 of Spanish Ministry of Science and Innovation. Francisco Morinha was supported by a Juan de la Cierva postdoctoral fellowship from Spain’s Ministry of Science and Innovation (FJCI-2017-32055). Support for the publication fee was provided by the CSIC Open Access Publication Support Initiative through its Unit of Information Resources for Research (URICI). The funders had no role in study design, data collection and analysis, decision to publish, or preparation of the manuscript.

==============================
The assessment of temporal variation in genetic features can be particularly informative on the factors behind demography and viability of wildlife populations and species. We used molecular methods to evaluate neutral genetic variation, relatedness, bottlenecks, and inbreeding in a declining population of Egyptian vulture (Neophron percnopterus) in central Spain. The results show that the genetic diversity remained relatively stable over a period of twelve years despite the decline in census and effective population sizes in the last decades. A relatively high proportion of nestlings from different and distant territories showed high relatedness in each study year. We also found support for an increasing impact of severe recent (contemporary) rather than distant (historical) past demographic bottlenecks, and the first evidence of inbred mating between full siblings coinciding with lethal malformations in offspring. The inbred nestling with feather malformations was positive to beak and feather disease virus recorded for the first time in this species. These results alert on recent and novel threats potentially affecting health and reducing the adaptive potential of individuals in this threatened species.

Introduction

The assessment of genetic diversity is crucial to understanding trends of wildlife populations and to implementing adequate strategies for conservation of endangered species (Willi, Van Buskirk & Hoffmann, 2006; Frankham, 2010). The evaluation of temporal variation in genetic diversity, bottlenecks, and relatedness can be particularly informative on the factors behind demography and viability of wild populations (Frankham, Ballou & Briscoe, 2002; Wootton & Pfister, 2013). This variation is often correlated with census size and population fragmentation depending on natural and anthropogenic threats (Frankham, Ballou & Briscoe, 2002; O’Grady et al., 2004). Both genetic and environmental factors also interact to determine individual health and fitness further contributing to demography and population dynamics (Keller et al., 2002; Reed, Briscoe & Frankham, 2002; Allentoft & O’Brien, 2010). During the last decades, anthropogenic activities have exerted increasing impacts on biodiversity worldwide. Among the main detrimental factors, disease is an increasing threat with anthropogenic change (Martin et al., 2010; Becker, Streicker & Altizer, 2015; Cunningham, Daszak & Wood, 2017) combined with low diversity impacting ability to fight parasites and pathogens (Spielman et al., 2004; Gupta, Robin & Dharmarajan, 2020).

Relatedness levels among individuals can be used as an indicator of genetic diversity in wildlife populations and species. The mating between close relatives can cause the loss of genetic variation due to random genetic drift, increasing the expression of recessive deleterious alleles, the loss of heterozygosity, and the extinction of functionally important alleles in the population (Keller & Waller, 2002; Charlesworth & Willis, 2009). Inbreeding in bird populations may result in several negative biological effects such as hatching failure, low offspring survival, and unviability due to physiological alterations and morphological malformations (Keller & Waller, 2002; Gupta, Robin & Dharmarajan, 2020). Therefore, inbreeding can influence the persistence of small and isolated populations by accelerating the extinction process in critically endangered species (Brook et al., 2002; Frankham, Ballou & Briscoe, 2002; O’Grady et al., 2006). This may arise especially if population decline is experienced quickly in recent history, rather than over long periods across the demographic history of species, thus respectively precluding and allowing the purging of deleterious alleles (Crnokrak & Barrett, 2002; Fox, Scheibly & Reed, 2008; Leberg & Firmin, 2008). These processes can be linked to recent and historic genetic bottlenecks differentially contributing to a progressive simultaneous reduction of genetic diversity and effective population size, and to the effects of inbreeding (Keller & Waller, 2002; Bouzat, 2010).

Multiple life-history traits and environmental factors can modulate the gain and loss of neutral genetic variation of populations and species (Lande, Enger & Sæther, 2003; Holderegger & Wagner, 2008). In long-lived species, this balance may be especially influenced by high maturity time, slow metabolism, and particular demographic, spatial, and social structure of populations, acting as buffer or promoting factors against population fluctuations and trends (Roff, 2002; Sæther et al., 2013). Therefore, any environmental and demographic alteration of these traits can increase stochasticity exerting pervasive effects on population size and genetic variation (Lande, Enger & Sæther, 2003; Palstra & Ruzzante, 2008; Jeppsson & Forslund, 2012). Among large long-lived birds, vultures provide a case study of certain extreme life history traits (Van Overveld et al., 2020), so they may be excellent models to study the impact of environmental and demographic stochasticity, and to test predictions on contemporary and historical variation in neutral genetic diversity, population trends, and extinction risk. Despite vultures being one of the most threatened groups of birds, the information on genetic diversity of populations and species is still limited (Godoy et al., 2004; Poulakakis et al., 2008; Martínez-Cruz & Camarena, 2018; Çakmak et al., 2019; Kleinhans & Willows-Munro, 2019; Davidović et al., 2020). As a consequence, there is a paucity of information on whether the interaction between life history traits, environmental features, and population characteristics can shape a genetic influence on the persistence and conservation of their populations. In particular, the spatial distribution of population nuclei interconnected by dispersing individuals contributing to gene flow is an important factor shaping patterns of genetic diversity throughout a species’ range (Holderegger & Wagner, 2008; Lowe & Allendorf, 2010), often associated to social behaviour and individual recognition of close relatives (Pusey & Wolf, 1996; Hatchwell, 2010). These factors can arise to minimize mean kinship within the population and prevent inbreeding, but no information is available on these mechanisms in vultures (Van Overveld et al., 2020). In addition, there is scarce knowledge on pathogens affecting vultures (Plaza, Blanco & Lambertucci, 2020) and no information on the interactions between genetic erosion, inbreeding, and disease. This knowledge is paramount to implement in-situ conservation management measures, and ex-situ actions such as captive breeding and reintroduction.

In this study, we used molecular methods to evaluate neutral genetic variation and relatedness in a declining population of Egyptian vulture (Neophron percnopterus) in central Spain. The main causes of the generalized decline of the Egyptian vulture includes illegal persecution through poisoning, habitat degradation, and reduction of food resources (BirdLife International, 2019). In addition, as consequence of the dependence on carrion of industrially-reared swine exploited at supplementary feeding stations across the study area (Blanco, 2014), this population has been recorded as highly exposed to pig-adapted Salmonella serotypes and to livestock antibiotics and resistant bacteria to these drugs (Blanco, 2018; Blanco, Junza & Barrón, 2017; Blanco & Díaz de Tuesta, 2021). Antibiotics ingested from livestock carcasses have been suggested behind immunosuppression, microbiota dysbiosis, and proliferation of opportunistic pathogens like yeast, filamentous fungi, and bacteria causing oral disease that can cause nestling mortality (Blanco, Junza & Barrón, 2017; Pitarch, Gil & Blanco, 2017; Pitarch, Gil & Blanco, 2020; Blanco et al., 2019). Previous studies have not highlighted the loss of genetic diversity as a main threat for the Egyptian vulture (BirdLife International, 2019) except in insular populations (Agudo et al., 2011b; Agudo et al., 2012).

As part of the long-term population monitoring, we recorded the identity of banded breeders to determine the nest of birth, natal dispersal, and reproductive success. To assess whether genetic diversity, relatedness, and inbreeding have changed in recent times, we compared these traits in two breeding seasons from the last two decades (2006 and 2018). In addition, we evaluated patterns and temporal variation in genetic signature of recent and historic demographic processes through bottleneck analysis. To further investigate inbreeding and its potential negative consequences, we examined the genetic relationships among nestlings fledged from a particular territory during a period of 15 years. Two of these nestlings subsequently mated, which allowed us to retrospectively investigate genetic relatedness and pedigree relationships between them and its influence in breeding performance and offspring health. This aimed to determine the extent of inbreeding as a relevant factor to be considered in the conservation management of this declining species.

Materials & Methods

Study species and study area

The Egyptian vulture is a small (∼2 kg) obligate scavenger living in a variety of open, arid, montane, and rugged landscapes in Southern Europe, Africa, and Asia (Cramp & Simmons, 1980; Del Hoyo, Elliott & Sargatal, 1994). Breeding adults nest on cliffs, defend territories against conspecifics, and behave as solitary or social foragers depending on the abundance, predictability, and concentration of food resources. Carcasses of wild animals (e.g., ungulates, lagomorphs, birds, and reptiles) and livestock represent the main food sources during the breeding season (Donázar, 1993; Blanco, Junza & Barrón, 2017; Blanco et al., 2019). Egyptian vultures are monogamous, highly philopatric, and territorial during the breeding season (Donázar, 1993; Sanz-Aguilar et al., 2017). This species shows a slow pace of life, with long generation time (14 years according to BirdLife International, 2019), average age of first breeding in the 7th year of age, low fecundity and reproductive rate (clutch size: 2 eggs; 0–2 fledglings per breeding attempt), and long lifespan (maximum age recorded in the wild: 24 years) (Sanz-Aguilar et al., 2017). This species is categorized globally as Endangered due to severe long-term declines through much of the distribution range (BirdLife International, 2019). The bulk of the European population is concentrated in Spain (about 1,500 pairs), where the breeding population has showed a strong decline in the last decades (BirdLife International, 2019).

The study was conducted in the province of Segovia (Castilla y Léon, Central Spain). This area holds a small and declining population of Egyptian vultures mostly nesting in cliffs located in two major systems of large gorges in two protected areas (Natural Parks of Hoces del Río Duratón and Hoces del Río Riaza) and their surroundings, while isolated pairs nest in small and scattered cliffs. The breeding population has suffered a sharp decline in the study area since the 1990s (from 48 pairs in 1993 to 27 pairs in 2019; Del Moral, 2009; Sanz-Aguilar et al., 2017). This population has been genetically characterized as forming part of the central-northern cluster from the distribution range in Iberia, including breeding sub-nuclei in Navarra and Aragón, north-eastern Spain (Agudo et al., 2011a). The Egyptian vulture pairs nesting in the area between Segovia province (the study area) and the northern distribution range in Iberia are also expected to belong to the same genetic cluster (Agudo et al., 2011a), as indicated by natal dispersal involving emigrants and immigrants recruited as breeders throughout this area (Serrano et al., 2021). Overall, for 19 Egyptian vultures recruited in or from the study area between 5 and 10 years after being marked as nestlings since 2003, we recorded a median dispersal distance of 41.2 km (range: 5.5–201). Most of the individuals recruited as breeders in the study area correspond to those also born in the study area (13 of 16). The remaining individuals were recruited in neighbouring provinces (Burgos and Soria respectively), while another individual was recruited in a more distant province (Zaragoza, Ebro Valley, north-eastern Spain). In addition, only two immigrant individuals born in the province of Zaragoza were recruited in the study area, while another individual born in Zaragoza was recruited in a neighbouring province (Guadalajara) to the study area. The study was carried out in accordance with the permission of the regional government of Castilla y Léon, Dirección General del Medio Natural, Servicio de Espacios Naturales.

Fieldwork and sampling procedures

An intensive monitoring program, including searching for territories, ringing nestlings, and identifying banded breeding individuals at territories was conducted at the Segovia province and its surroundings during the last four decades (Martinez & Blanco, 2002; Sanz-Aguilar et al., 2017). The nests were accessed by climbing when nestlings were feathered (at ages of about 50 days) but without risk of leaving the nests. The clutch size in this species is of two eggs, but in a proportion of nests only one nestling survives, mostly the first-hatched nestling (Donázar et al., 2020). Therefore, sampled nestlings can correspond to the single nestling in the nest (brood size = 1) or two siblings in the same nest (brood size = 2). Nestlings (n = 180) were measured, weighed, and banded with metal and plastic rings with alphanumeric codes allowing individual identification at long distance with terrestrial telescopes. A sample of blood as source of DNA was taken from the brachial wing vein and preserved in absolute ethanol.

To evaluate temporal trends in genetic relatedness, bottlenecks, and inbreeding in the population, we selected nestlings sampled in 2006 (n = 19 from 15 territories) and 2018 (n = 14 from 10 territories). The sampled nestlings correspond to about 40–50% of the active territories each year in the population, as a variable proportion of breeding pairs failed in the reproduction each year (Martinez & Blanco, 2002; Sanz-Aguilar et al., 2017). Each territory can include several nesting sites, but only a particular nesting site is used each year in each territory. Therefore, we sampled a relatively well-represented proportion of the territories available for nestling sampling each study year. Overall, our study includes the sampling of 42 Egyptian vulture individuals (see Appendix S1).

In the breeding season of 2017, we located a novel breeding pair composed of two individuals born in the same nest but in different years. Because breeders can use the same nest and territory year to year (Sanz-Aguilar et al., 2017), we assessed whether these individuals were genetically related or if they simply born in the same territory used by different breeders in different years. To this aim, we evaluated the relatedness among the nestlings (n = 10) born in 2004 (n = 1), 2005 (n = 1), 2006 (n = 1), 2007 (n = 2), 2010 (n = 2), 2018 (n = 2) and 2020 (n = 1) in the two nesting sites (distant <200 m) included in this single territory (a single nesting site used each year). This allowed us to evaluate the validity of the relatedness estimates in reflecting full-sib relationships. To determine whether the same breeders used the same territory across the years, we collected feathers of the breeders attending this particular territory during the nest access to band the nestlings in 2018. We wanted to determine whether these breeders were the parents of the nestlings raised in the same territory the previous years, and then to establish long-term nest re-use and changes in the composition of the pair mate.

DNA extraction, molecular sexing and microsatellite genotyping

The DNA was isolated by using the Quick-DNA Miniprep Plus Kit (Zymo Research) according to manufacturer’s protocols. Molecular sexing was performed using specific primers for partial amplification of the CHD1 gene (Fridolfsson & Ellegren, 1999).

Seventeen microsatellite markers previously characterized for Egyptian vulture were analysed: BV9, BV13, BV14, BV20, NP39, NP51, NP93, NP140, NP141, NP155, NP163, NP166, NP229, NP249, NP257, NP259, and NP296 (Kretzmann et al., 2003; Agudo et al., 2008). Microsatellite forward primers were labelled with 6-FAM, NED, PET, and VIC fluorescent dyes to determine the genotypes using capillary electrophoresis. PCR amplifications were performed in a total volume of 10 µl containing 5 µl of 2x MyTaq HS Mix (Bioline), 2 µM of each primer, and 5 ng DNA. PCR thermal conditions were as follows: initial denaturation at 95 °C for 5 min, followed by 40 cycles of 95 °C for 30 s, 58 °C for 1 min, 72 °C for 30 s, and a final extension at 60 °C for 10 min. Amplified fragments were electrophoresed on an ABI PRISM 3130xl Genetic Analyzer (Applied Biosystems) using the GeneScan 500 ROX size standard. Allele sizes were determined using Peak Scanner v.3.0 (Thermo Fisher Cloud) and are presented as supplementary data (Appendix S1).

Evaluation of microsatellite loci quality and genetic diversity

The screening of genotyping errors, large allele dropout, null alleles, and stuttering were performed with MICRO-CHECKER v.2.2.3 (Van Oosterhout et al., 2004) using Bonferroni (Dunn-Sidak) adjusted confidence intervals (95%) obtained after 10,000 Monte Carlo simulations. The significance of linkage disequilibrium (LD) and deviations from Hardy-Weinberg equilibrium were assessed using GENEPOP v.4.4.2 (Rousset, 2008) applying the Markov chain method (10,000 dememorisation steps, 1000 batches and 5000 iterations/batch), and a sequential Bonferroni correction. The number of alleles per locus (NA), observed and expected heterozygosities (HO and HE) were calculated with GENALEX v.6.5 (Peakall & Smouse, 2012). Allelic richness (AR) values were obtained using FSTAT v.2.9.3.2 (Goudet, 1995).

Analysis of genetic differentiation

The possible genetic differentiation between offspring from 2006 and 2018 was assessed using the FST estimator of Weir & Cockerham (1984), as implemented in GENALEX v.6.5. Allele frequency differentiation between samples of 2006 and 2018 was tested using Fisher’s exact test in GENEPOP v.4.4.2. We carried out a multi-locus spatial autocorrelation analysis (Smouse & Peakall, 1999) for both sampling years pooled and independently by using GENALEX. The spatial autocorrelation coefficient of genetic distance (r) was estimated for a maximum of 9 distance classes (a total distance of 90 km). Significance tests for each distance class were performed using 9,999 random permutations and 9,999 bootstrap replicates. In addition, we performed a Bayesian clustering analysis using the program STRUCTURE v.2.3.4 (Pritchard, Stephens & Donnelly, 2000). The analysis consisted of 10 independent runs for K values ranging from 1 to 3, each with 1,000,000 MCMC iterations after a burn-in of 100,000 interactions, under a model of admixture and correlated allele frequencies. The outputs were processed in the CLUMPAK server (Kopelman et al., 2015).

Genetic relatedness and inbreeding measures

Various studies have been reported that genome-wide single nucleotide polymorphisms (SNPs) may outperform microsatellite markers in kingship assignment and relatedness estimations (e.g., Hauser et al., 2011; Weinman, Solomon & Rubenstein, 2015; Thrasher et al., 2018). However, the microsatellites could be also a reliable choice if (i) there is a microsatellite panel available for the species under study, (ii) the markers have suitable polymorphic levels, (iii) they have a good resolution power and (iv) DNA samples have low-quality to generate reliable genome wide SNPs (Flanagan & Jones, 2019). We selected a well-known microsatellite panel previously developed and applied in population genetic studies of Egyptian vultures for our relatedness approaches, taking into account all these factors, our target species and the biological samples.

Relatedness (r) among individuals was calculated using different estimators to evaluate the robustness and reliability of our inferences, since the performance of the different relatedness algorithms are dependent of the dataset used (Anderson & Weir, 2007; Wang, 2014). The COANCESTRY application (Wang, 2011) was used to test the performance of five relatedness estimators reported by Wang (2002), Lynch & Li (Lynch, 1988; Li, Weeks & Chakravarti, 1993), Lynch & Ritland (1999), Ritland (1996) and Queller & Goodnight (1989). The pairwise relatedness values (dyads) were calculated and the best estimators were evaluated using the reference values obtained for the know full-sib relationships (full-sibs 1V5, 1V9 and G15749; full-sibs 290, 291, 32U, 32T, 9MA and 9LX; full-sibs 9MC and 9MF; see Appendix S1). The mean pairwise relatedness estimated from these individuals were compared with the theoretically expected value for full-siblings (0.5). Relatedness were also estimated for the group of all Egyptian vultures included in the study (a value less than 0.5 is expected). GENALEX v.6.5 (Smouse & Peakall, 2012) was applied to generate the tridiagonal matrix of pairwise estimates of relatedness using the Queller and Goodnight’s estimator (QG-r), which minimizes a downward bias for small sample sizes (Queller & Goodnight, 1989). The ML-RELATE software (Kalinowski, Wagner & Taper, 2006) was used to estimate the genetic relationships between all individual vultures. This program calculates the maximum likelihood estimates of relatedness (ML-r) and relationship categories between pairs of individuals from genotypic data. The likelihood values of the relationships available from the software (U, Unrelated; HS, Half sib; FS, Full sib; PO, Parent-Offspring) were calculated to determine the relationship with highest likelihood for each pair of individuals. When individuals born in the same year were categorized as PO, kinship was assumed to be FS. The inbreeding coefficients (FIS) were estimated using GENEPOP v.4.4.2 (Rousset, 2008) to assess the level of inbreeding in the groups of individuals analysed. The program CERNICALÍN v.1 (Aparicio, Ortego & Cordero, 2006) was used to calculate the homozygosity by loci (HL), estimating the indices of inbreeding considering the proportion of loci within the genotypes of an individual that are homozygous. Statistical differences between years in the parameters of genetic diversity, relatedness, and inbreeding were assessed by Mann–Whitney U-test.

The spatial pattern of relatedness was tested by two-tailed Mantel test with 9,999 permutations considering the geographic distances (in km) between nests (Appendix S2) and the QG-r and ML-r values of each nestling from 2006 and 2018 (a single nestling per nest selected at random was considered in the case of broods with two siblings to avoid pseudoreplication).

Demographic events and effective population sizes

The detection of potential bottleneck events that lead to severe population size reductions in the recent (contemporary) past over the last decades was performed using the program BOTTLENECK version 1.2.02 (Piry, Luikart & Cornuet, 1999). The stepwise mutation model (SMM), infinite allele model (IAM), and two-phase mutation (TPM) model and mode-shift test were used. This analysis was complemented with the M-ratio model implemented in the software M_P_VAL (Garza & Williamson, 2001), which allows the detection of bottlenecks that occurred in the recent and distant (historical) past over centuries or millennia (Piry, Luikart & Cornuet, 1999; Williamson-Natesan, 2005). All parameters used in the analysis were previously described (see Blanco et al., 2021).

The contemporary effective population size (Ne) was estimated using the program LDNE v.1.31 (Waples & Do, 2008). The random mating model was used. Alleles with frequencies lower than 0.02 were excluded and the confidence intervals were obtained using jack-knife statistics.

Screening of infectious agents

The inbred offspring (nestling with feather malformations, ref: 9MC from 2018; and nestling apparently healthy, ref: 9MF from 2020) were analysed for the presence of three potential infectious agents that can cause plumage alterations (beak and feather disease virus, BFDV; avian polyomavirus, APV; and Salmonella). Specific molecular markers previously reported were used in the PCR screening of BFDV (Amery-Gale et al., 2017) and APV (Johne & Müller, 1998). PCR was performed in a reaction mixture of 10 µl containing 5 µl of 2x MyTaq HS Mix (Bioline), 2.5 µM of each primer and ∼20 ng of template DNA. The amplification protocol was composed of the following steps: 95 °C for 5 min followed by 40 cycles of 95 °C for 30 s, annealing at 58 °C for 1 min, 72 °C for 30 s, and a final extension at 60 °C for 10 min. To determine the presence of Salmonella, nestlings were sampled for cloacal mucosa using sterile microbiological swabs with Amies transport medium. The samples were analysed by conventional microbiological culture (ISO 6579:2002/Amd. 1:2007) and by using the real-time PCR IQ-CHECK™ Salmonella II kit (Bio-Rad) and Easy I protocol, according to the manufacturer’s instructions (see details of both methods in Blanco & Díaz de Tuesta, 2018; Blanco & Díaz de Tuesta, 2021).

Ethics statement

Our study followed ethical guidelines proposed for the Spanish Royal Decree 1205/2005 on the protection of animals used in experiments and scientific research. The study was carried out in accordance with permits from the Spanish Bird Ringing Centre (Permit Number: 530115), and the regional government of Castilla y Léon (Expte: EP/CyL/298/2016).

Results

Genetic data quality, genetic diversity and differentiation

The probability of null alleles was negligible for all loci except for NP140 (dataset of all individuals and offspring 2018). Therefore, the subsequent analyses were carried out with and without this locus. The results were similar when NP140 was included and excluded from the data set. Therefore, we present the results including this locus. Only the loci NP140 (offspring 2018) showed a significant deviation from Hardy-Weinberg equilibrium (HWE) after Bonferroni correction. When all individuals were analysed in the same dataset no significant deviation from HWE was observed (Table S1). Exact tests for genotypic LD confirmed the absence of physical linkage at most loci. Only two significant linkages were found; however, these significances were lost when all siblings were removed from the dataset. In this context, all microsatellite loci were included in the multilocus analysis.

All microsatellite loci were polymorphic, with a total of 75 different alleles amplified. The number of alleles per locus ranged from two (BV14, BV20 and NP257) to eight (NP39), with an average of four alleles (Table S1). In the offspring of 2018, only one allele for the locus NP51was detected. Mean values of AR, HO, and HE did not differ between nestlings from 2006 and 2018 (Table 1).

Table 1 Mean ± SD values of genetic diversity, relatedness, and inbreeding of nestling Egyptian vultures sampled in 2006 and 2018, and both years pooled in Segovia, central Spain.

	2006 (n)	2018 (n)	Difference between years z (p)	Total pooled (n)	
Genetic diversity					
AR	3.81 ± 1.37 (17)	3.71 ± 1.72 (17)	0.00 (1.00)	3.76 ± 1.53 (34)	
HO	0.62 ± 0.22 (17)	0.57 ± 0.25 (17)	0.03 (0.98)	0.59 ± 0.23 (34)	
HE	0.58 ± 0.18 (17)	0.57 ± 0.23 (17)	−0.05 (0.96)	0.57 ± 0.20 (34)	
Relatedness and inbreeding				
QG-r	−0.01 ± 0.18 (105)	−0.04 ± 0.22 (45)	1.07 (0.28)	−0.02 ± 0.19 (150)	
ML-r	0.07 ± 0.09 (105)	0.08 ± 0.11 (45)	−0.09 (0.93)	0.07 ± 0.10 (150)	
FIS	−0.04 ± 0.19 (17)	0.01 ± 0.31 (15)	−0.43 (0.67)	0.02 ± 0.25 (32)	
HL	0.33 ± 0.09 (15)	0.39 ± 0.10 (10)	−1.36 (0.17)	0.35 ± 0.10 (25)	
Notes.

Genetic diversity was assessed by allelic richness (AR), observed heterozygosity (HO), and expected heterozygosity (HE). Relatedness and inbreeding were assessed through Queller and Goodnight’s pairwise relatedness values (QG- r), maximum likelihood estimates of relatedness (ML- r), inbreeding coefficients (FIS), and homozygosity by loci (HL). n represents sample size (AR, HO, HE and FIS—number of markers considered in the estimations; QG- r and ML- r number of pairwise comparisons; HL—number of individuals). Statistical differences between years were assessed by Mann-Whitney U-test.

Genetic differentiation was not significant between offspring from 2006 and 2018 (FST = 0.02, p = 0.38). No significant differences were detected when Fisher’s exact tests of genetic differentiation were calculated from allele frequency data (Table S2). We found no evidence of genetic spatial autocorrelation when all individuals were analysed together and when the offspring of 2006 and 2018 were analysed as independent groups (Fig. S1). The results of Bayesian clustering analysis also show no evidence of genetic structure and differentiation between years (Fig. S2).

Relatedness

The evaluation of the five genetic relatedness estimators showed a good performance for most of the estimators (Table S3; Appendix S3). The estimators that allowed a better identification of the true full-sibs using the relatedness values were Wang (0.59 ± 0.13), Lynch & Li (0.60 ± 0.12) and Queller & Goodnight (0.60 ± 0.12) (Table S3). The mean relatedness values when all individuals were analysed were also very similar for all estimators, ranging from −0.02 ± 0.24 (Queller & Goodnight) to −0.01 ± 0.26 (Wang and Lynch & Li) (Table S3). In addition, most of the pairwise values obtained for the five estimators were highly correlated (average correlation value of 0.87 ± 0.07) (see Table S4 for pairwise correlation values). Therefore, for simplification purposes, we showed only the relatedness values for Queller & Goodnight estimates (QG-r) in the following analysis.

Measures of QG-r (Appendix S4) and ML-r (Appendix S5) were highly correlated (Mantel test: r = 0.835; p < 0.0001). The strategy used for categorizing the genetic relationships allowed the accurate assignment of all direct relationships in individuals with known pedigree, and only two deviations in the kinship assignation were observed for inferred relationships. Overall, we correctly assigned 93% (26 of the 28 pair-wise relationships) obtained for the group of individuals with known pedigree. Therefore, it is expected that relationship categories estimated for individuals with unknown pedigree reflect the actual relatedness with accuracy. For nestlings from 2006, 33 half-sib (19.8%) and three full-sib (1.8%) relationships were estimated, excluding the four full-sibs from broods of two nestlings reared in the same nest and year (Appendix S6). For nestlings from 2018, 13 half-sibs (14.9%) and four full-sibs (4.6%) relationships were estimated, excluding the four full-sibs from the same nest (Fig. 1A; Appendix S6). The geographic location of the territories sampled in 2018 is shown in Fig. 1B. The proportion of nestlings in each kinship category did not differ between years (Fisher exact test, p = 0.19), being about 20% those estimated being closely related individuals (half and full-sibs) when pooled years; for this analysis, we considered a single nestling selected at random from broods of two nestlings to avoid pseudoreplication. Mean values of relatedness and inbreeding were similar between nestlings from 2006 and 2018 (Table 1). The HL values for offspring of 2006 ranged from 0.19 to 0.52 (mean = 0.33 ± 0.09), while these values were slightly higher (mean = 0.39 ± 0.10) in offspring from 2018 (range = 0.26–0.59), indicating that on average 35–50% of an individuals’ loci are homozygous (Table 1; Table S5).

Figure 1 Inferred relationships and geographic distribution of the offspring in 2018.

(A) Dyads of closely related Egyptian vulture nestlings from different territories sampled in Segovia, central Spain; the same colours of symbols represented together (circles represent females and squares represent males) were used for broods of two nestlings. (B) Geographic location of the sampled territories; the colours representing each territory are the same than those representing nestlings from these territories. The arrow indicates the dispersal from the natal to the breeding territory of an inbred pair formed of full-sibling from the same territory but different cohorts (see Fig. 3 for the genealogy of this familiar lineage).

All territories showed at least one closely related individual (full or half sib) among the nestlings sampled in 2006 (100%, n = 15 territories), and most territories in 2018 (90%, n = 10). The number of territories with closely related nestlings ranged from one to seven in 2006, and from zero to four in 2018 (Fig. 2; see also Figs. 1A and 1B for the specific and geographical representations of these relationships in 2018). Relatedness between nestlings from different nests (Fig. 1A for 2018) showed no apparent spatial pattern (Fig.‘1B), as inter-nest distance was not significantly correlated with QG-r and ML-r values of each nestling from 2006 (r =  − 0.06, p = 0.62, n = 105; r =  − 0.08, p = 0.43, n = 105, respectively) and 2018 (r = 0.13, p = 0.35, n = 45; r = 0.08, p = 0.57, n = 45, respectively).

Figure 2 Frequency (%) of territories with closely related (half- and full-sibs) nestling Egyptian vultures.

The data are represented attending to the number of territories with closely related nestlings in each study year. The specific and geographic representations of these relationships are shown in Fig. 1A and Fig. 1B, respectively, for nestlings from 2018. The number of territories (n) sampled each year is shown.

Demographic history and effective population sizes

The analysis with BOTTLENECK suggests population contractions in the recent past by analysing the samples from 2006 and 2018 populations (Table 2). Evidence of recent bottlenecks was found through SMM and the mode-shift test in 2006 and through all models in 2018 (Table 2). The M-ratio test does not support the occurrence of bottleneck events in a more distant (historical) past (values of M >Mc) (Table S6). Similar results were attained when nestlings from different territories and years were pooled to increase sample size (Table 2; Table S6).

Figure 3 Pedigree of the Egyptian vulture familiar group analysed.

Circles represent females and squares males, including an inbred pair formed by full-siblings from different cohorts (ref. 291-32U) and their offspring. Double arrows represent mated pairs. Years are shown within orange rectangles. Dashed lines represent dyads with relatedness estimates corresponding to parent-offspring values, but actually corresponding to grandparents and grandchildren, the last being the offspring of the inbreed pair. All relationship categories between dyads of nestlings correspond to actual sibship, as determined by pedigree and confirmed from genotypic data. The estimate as unrelated of the unmarked progenitors of all nestlings except those from the inbreed pair was determined from genotypic data. The occurrence or absence of PBFV and malformations are shown for the offspring of the inbred pair.

Table 2 Results of the bottleneck analysis.

P-values of the one-tail Wilcoxon sign-rank tests for heterozygote excess are shown for the Two-Phase Model (TPM), Stepwise Mutation Model (SMM), and Infinite Allele Model (IAM). For the Mode-shift test, modes obtained for each group are indicated.

Year	n	TPM P-values	SMM P-values	IAM P-values	Mode-shift	
2006	15	0.040	0.290	0.008	Normal L-shaped	
2018	10	0.003	0.017	<0.001	Shifted mode	
2006 + 2018*	21	0.008	0.270	<0.001	Normal L-shaped	
Notes.

* Only individuals from different territories.

Estimates of effective population size (Ne) for each sampled year were 100 (CI = 24-inf) for 2006 and 48 (CI = 13-inf) for 2018. The value of Ne considering all sampled individuals (2006 + 2018) was 53 (CI = 26–314).

Long-term genetic relationships and inbreeding effects in a familiar lineage

The retrospective analysis of relatedness in the sampled familiar linage showed a replacement of the mated pair present in 2004–2006 by another pair in 2007–2018 in this particular territory. This was confirmed by several facts: (i) nestlings from 2004–2006 were full sibs, and therefore shared the same progenitors; (ii) nestlings from 2007, 2010, and 2018 were also full sibs (Fig. 3) but unrelated to nestlings from 2004–2006 except for a single half-sib relationships, and therefore were offspring of a different breeding pair; (iii) nestlings from 2007–2018 showed parent–offspring relationships with their putative parents sampled for feathers in 2018 (Fig. 3); (iv) nestlings from 2004–2006 were unrelated with the female breeder but showed half-sib relationships with the male breeder in 2018 (Appendix S6).

Mean relatedness values for the familiar group from 2007 to 2018 was high (QG-r = 0.62 ± 0.11; ML-r = 0.53 ± 0.12), as is expected for dyads of closely related individuals. Two of these full siblings mated between them (QG-r = 0.59; ML-r = 0.43) in 2017: a female (ref. 291) born in 2007 and a male (ref. 32U) born in 2010 (Fig. 3), and nested in a territory distant 59 km from their natal nest (Fig. 1B). We found two half-sib relationships in this familiar lineage (Fig. 3): (i) between the male breeder (ref. M) and his granddaughter (ref. 9MC) from the inbred pair (refs. 291-32U); (ii) between a nestling from 2017 (ref. 290) and the son (ref. 9MF) of the inbred pair (refs. 291-32U). A single half-sib relationship was recorded by molecular methods between two actually full-sibs from different cohorts in this territory, which suggest slight deviations in the kinship assignation due to allelic frequency. Relatedness of all offspring with their actual parents and grandfathers were confirmed by pedigree and molecular methods. Note, however, that the ML-RELATE software may not differentiate true full-sibs and parent–offspring relationships in familiar cases with many closely related individuals, but these relationships were ascertained in individuals with known pedigree in this familiar lineage.

The inbred pair failed to reproduce in 2017. In 2018, the same pair reared a female offspring (ref. 9MC) that was confirmed to be daughter of their putative parents by microsatellite allelic patterns (Appendix S6) and relatedness values (QG-r = 0.72 ± 0.18; ML-r = 0.64 ± 0.19). This nestling showed alterations in feather structure and appearance across all the plumage (Fig. 4A). These acute malformations mainly consisted in greasy plumage, constricted calamus in all flight feathers with barbs appearing only in the distal part, and pinched appearance in a proportion of feathers (Fig. 4B). As a consequence of these severe and generalized malformations, this individual was unable to fly, and was retired by competent authorities and admitted in a wildlife recovery centre. In 2019, the same inbred pair failed in the reproduction apparently due to nestling predation. In 2020, the same pair produced an apparently healthy offspring (ref. 9MF) that was confirmed as son of their putative parents by microsatellite allelic patterns (Appendix S6) and relatedness values (QG-r = 0.72 ± 0.07; ML-r = 0.60 ± 0.07). This nestling was confirmed as full-sib of the nestling with feather malformations in 2018 (QG-r = 0.73; ML-r = 0.64), and as full-sibs with all other nestlings from this family lineage in 2007–2018 (Fig. 3). Values of HL were high for both inbreed individuals (Ref. 9MC = 0.48; Ref. 9MF = 0.54) as expected for offspring from parental full-sibs (Table S5).

Figure 4 Fledgling Egyptian vulture with feather malformations, daughter of an inbred pair formed by full siblings.

(A) Alterations in feather structure and appearance across all the plumage. (B) Detail of the constricted calamus in all flight feathers with barbs appearing only in the distal part. Pictures: G Blanco.

The screenings of infectious agents performed in these inbreed offspring showed a BFDV positive result in the nestling with malformations (Ref. 9MC) and negative results for APV and Salmonella. All results were negative in the nestling apparently healthy (Ref. 9MF).

Discussion

Genetic diversity, relatedness and inbreeding

A growing body of evidence has related reduction in genetic diversity with declining population size of wild species (Frankham, 1996; Willoughby et al., 2015). Our data show that the genetic diversity in a small open population of Egyptian vulture remained relatively stable despite of its decline in census and effective population size in the last decades. A relatively high proportion of nestlings from different and distant territories showed high relatedness in each study year. In addition, we found an increasing impact of recent demographic bottlenecks and the first evidence of mating between full siblings coinciding with lethal malformations in offspring.

We found similar levels of genetic variability and non-significant differences between two breeding seasons twelve years apart. This lack of inter-year differences is not surprising considering that the time period between both study years (12 years) is shorter that the generation time in the Egyptian vulture (14 years; BirdLife International, 2019). Some studies have suggested that FIS values tend to be lower (more outbred) when estimates are calculated using only samples from juveniles, and higher (more inbred) in samples from adults (Basset, Ballouxf & Perrin, 2001; Parreira & Chikhi, 2015). This is a potential consequence of a heterozygote excess increased by reproductive events, which are later counteracted by dispersal (Basset, Ballouxf & Perrin, 2001). Thus, we cannot exclude a potential underestimation bias in our estimates of inbreeding coefficients, since they were obtained from nestlings. Overall, the values of genetic diversity estimates were similar to those reported for several declining vulture species (Gautschi et al., 2003; Arshad et al., 2009; Çakmak et al., 2019), higher than those observed in other declining species (Kleinhans & Willows-Munro, 2019), and even higher than in vulture species showing larger population size and stable or increasing demographic trends (Le Gouar et al., 2008; Arshad et al., 2009). Therefore, these results should be interpreted with caution because different markers were used to estimate genetic diversity indices within and among species (see Väli et al., 2008). In addition, low genetic diversity is not always the result of a reduction in effective population size, as the former can be high in species with very small, isolated, and declining populations (Kekkonen, Wikström & Brommer, 2012; Woolaver et al., 2013). A low genetic diversity may be due to demographic bottlenecks and population rebounds, as well as due to founder events with a reduced number of individuals, and absence of random mating (Thévenon et al., 2003; Kekkonen, Wikström & Brommer, 2012). Current levels of neutral genetic diversity, relatedness, and inbreeding are similar to those reported for this species in the study area and other regions in continental Spain using the same suite of microsatellite loci (Agudo et al., 2011a). However, the values reported previously for the study area are not strictly comparable with those presented here because the former partially include full-grown individuals of unknown geographical origin (natal nest), age, and cohort. These individuals can include immigrants from other genetic clusters and geographical areas due to natal dispersal and non-breeding nomadic movements (IBERIS, 2007; Sanz-Aguilar et al., 2017).

Our results show a relatively high proportion (about 20%) of dyads comprising half- and full-sibs relationships among nestlings born in different territories in each study year. Indeed, a proportion of nestlings from distant territories in each year were genetic full-siblings. However, we found no evidence of a relationship between pairwise geographic inter-territory distances and pairwise genetic relatedness between nestlings from different territories in each cohort. This suggests a low contribution of extra-pair paternity to these intra- and inter-nest kinship relationships. Although extra-pair copulation has been reported at low frequency in the Egyptian vulture (Donázar, Ceballos & Tella, 1994), the contribution to extra-pair paternity remains generally unexplored in vultures (Le Gouar et al., 2011). Our results showed exclusive full-sib relationships between nestlings from the same nest and cohort (broods of two siblings). However, we found two half-sib relationships in the familiar lineage with known genealogy. This can be indicative of slight and conservative deviations in the kinship assignation due to allelic frequency, which deserves further assessment including more parents and offspring with known pedigree. The assessment of genetic relatedness using genome-wide SNP loci could improve the estimates (Flanagan & Jones, 2019), although our microsatellite panel with a probability of an accurate kinship assignment higher than 90% seems reliable enough for the aims of this study. Proportions of dyads falling in the kinship categories were higher than those recorded in other vulture species, although these relatedness estimates were calculated for wild and captive full-grown individuals (Gautschi et al., 2003; Ishtiaq et al., 2015). Given that these studies analysed relatedness of individuals of different ages, they can include actual parent–offspring relationships and full and half siblings from different cohorts. Instead, we found closely related individuals among nestlings from the same cohorts but distant territories, which clearly indicate that their parents were also closely related within and between pair mates from different territories. At the scale of the whole population, we found that most territories showed closely related individuals among the nestlings sampled in each study year. Indeed, nestlings of each particular territory were closely related to those from a variable number of other territories in each year. For instance, nestlings from three territories were closely related to those from other seven territories each in 2006, when the sampling was more extensive, while nestlings from one territory were closely related to those from up four territories in 2018. This implies that each one of these nestlings was closely related to those from about half of the territories sampled each year. Given that we sampled a representative proportion of the successful pairs each year, this further indicates a high relatedness in the population. Overall, this highlights the importance of knowing natal origin, age, and cohort in the assessment of genetic diversity and relatedness of wildlife populations.

Demographic history

The analysis of bottlenecks detected a clear signature of genetic contractions during the last decades, coupled with the recent decline according to census and effective population size in the study area and other regions included in the same genetic cluster (Carrete et al., 2007; Agudo et al., 2011a; Sanz-Aguilar et al., 2017). The significance observed in the heterozygosity excess contrasts with the consistent and non-significant results of the M-ratio test in both study years, which indicate that historical bottlenecks for the last centuries or millennia are not evident (Williamson-Natesan, 2005; Peery et al., 2012). However, it is important to highlight that statistical power of bottlenecks tests is highly related to the number of samples and markers used, which in many cases affect the detection of historical population reductions (Peery et al., 2012; Hoban, Gaggiotti & Bertorelle, 2013). Furthermore, the different sensitivity of heterozygosity-excess and M-ratio tests to possible violations of mutation model assumptions may affect the inferences about the timing of population declines (Peery et al., 2012). Therefore, we cannot completely rule out the occurrence of historical bottlenecks (distant past) in this population. The clear signature of recent bottlenecks disagrees with the relatively high values of genetic diversity estimates. This apparent contradiction may arise because of long generation times owing to delayed breeding in this long-lived species (Sanz-Aguilar et al., 2017) can still reflect the retained genetic diversity in the parents of the nestlings analysed. This time lag associated to a slow pace of life could act as an intrinsic buffer to stave off genetic erosion in long-lived species (Hailer et al., 2006). This implies that long-term studies are necessary to adequately detect temporal changes in genetic diversity in these species. Given that Egyptian vultures can reproduce during many consecutive years across their long lifespans (Sanz-Aguilar et al., 2017), some of these breeders can be old enough to still harbour relatively high genetic diversity typical of not bottlenecked populations. In support of this possibility, no evidence of recent bottlenecks was found in the study area sampled prior to our study (TPM model, p = 0.153, from Agudo et al., 2011a). This suggests that the impact of recent bottlenecks has increased in the last two decades to the point that it can be observed in its current progression in parallel to the reduction of the effective and census population size, but still without a clear reflection in genetic diversity estimates (see also Çakmak et al., 2019). However, the interpretation of these estimates should be made with caution when the proportion of closely related individuals in the population is high (Gautschi et al., 2003). In addition, the values of relatedness previously reported for the study area were obtained by pooling data from local nestlings with full-grown individuals of unknown natal origin, age, and cohort captured in foraging areas (IBERIS, 2007; Agudo et al., 2011a). Therefore, the lack of recent bottlenecks previously to our sampling could arise through an increase in genetic diversity by including individuals from other geographic regions and genetic clusters in the relatively continuous distribution range across central-north Iberia (Agudo et al., 2011a). This rationale could also apply to our data because, although the nestlings were sampled in a relatively small area of the distribution range of this genetic cluster, a small proportion of breeders could have been born outside the study area. In this case, the evidence of recent bottlenecks is supported by our conservative data, potentially including a proportion of offspring from immigrant individuals, which should be confirmed with a more extensive assessment over a larger geographical area.

Implications of inbreeding and emerging pathogens in conservation

The increase of close kin among mated breeders in small populations can lead to inbreeding depression. The consequences of a reduction of genetic diversity on offspring health and fitness can be severe for the affected populations (Charlesworth & Willis, 2009). This is a critical factor in the conservation of endangered species when it occurs rapidly enough to preclude the purging of deleterious alleles (Brook et al., 2002; Fox, Scheibly & Reed, 2008). Consequently, host susceptibility to disease agents is expected to increase due to inbreeding, although their effects on health can surface depending on whether the population is exposed to particularly dangerous and contagious pathogens (Smith, Sax & Lafferty, 2006; Gupta, Robin & Dharmarajan, 2020). The evidence of recent bottlenecks and the reduction in census and effective population size coincided in time with the first evidence in this species of close inbreeding, confirmed by pedigree of individually-marked nestlings monitored subsequently, and by genetic analysis. The inbred pair showed low reproductive success (only one apparently healthy nestling raised from four reproductive attempts) and the first evidence in this species of lethal feather malformations in offspring, which suggests a possible cause-and-effect relationship. The affected nestling showed alterations in the overall plumage that precluded fledging and these malformations remained the following two years in captivity. This kind of feather dystrophy resembles those characterizing the pinching off syndrome reported associated to inbreeding in several raptor species in absence of any identified aetiological agent (Müller et al., 2007). Similar malformations have been recorded in parrot and allies (Psittaciformes) affected by the beak and feather disease virus (BFDV), and recently this avian circovirus has been recorded affecting non-psittacine species, especially in small and isolated populations and species with low genetic diversity (Raidal & Peters, 2018). We found that the nestling with generalized feather malformations was positive for this virus, which genetic sequence is currently being characterized to determine its phylogenetic position informing on its potential origin (Harkins et al., 2014; Morinha et al., 2020). In contrast, the apparently healthy sibling (born in 2020) of the affected offspring (born in 2018) from the same inbred pair was negative to BFDV. This suggests a potential interaction between inbreeding and malformation depending on infection by BFDV, although other non-surveyed pathogens or pollutants could be involved in disease causation, which requires further research. In support of an interaction between inbreeding and malformation due to BFDV, we found other BFDV-positive nestlings in the population (n = 12), but no other with malformations has been recorded in previous studies (Blanco, Junza & Barrón, 2017; Blanco et al., 2020; Sanz-Aguilar et al., 2017).

Conclusions

Our study highlights on high relatedness, recent demographic bottlenecks, and inbreeding as concerning and increasing threats potentially reducing the adaptive potential of this declining population. This may arise directly through the effects of genetic erosion on health and fitness, and in combination with the detrimental effects of food-born pharmaceuticals and pathogens, including BFDV recorded for the first time in this species. Further research is needed to understand the relationships between demographic bottlenecks, inbreeding, susceptibility to pathogens, and health in the Egyptian vulture and other species of conservation concern.

Supplemental Information

Supplemental Information 1 Supplemental data

Click here for additional data file.

Appendix S1 Genotypes of each individual

Click here for additional data file.

Appendix S2 Matrices of geographic distances between nests

Click here for additional data file.

Appendix S3 COANCESTRY pairwise relatedness values

Click here for additional data file.

Appendix S4 Queller and Goodnight pairwise relatedness values

Click here for additional data file.

Appendix S5 Maximum Likelihood relatedness values

Click here for additional data file.

Appendix S6 Matrix of relationships

Click here for additional data file.

We thank JL González del Barrio, Ó Frías, B Arroyo, F Martínez, JP Calle and A González for their help with fieldwork, and JA Donázar for providing the microsatellite markers.

Additional Information and Declarations

Competing Interests

Author Contributions

Animal Ethics

Field Study Permissions

Data Availability

The authors declare there are no competing interests.

Guillermo Blanco conceived and designed the experiments, analyzed the data, prepared figures and/or tables, authored or reviewed drafts of the paper, fieldwork, and approved the final draft.

Francisco Morinha conceived and designed the experiments, performed the experiments, analyzed the data, prepared figures and/or tables, authored or reviewed drafts of the paper, and approved the final draft.

The following information was supplied relating to ethical approvals (i.e., approving body and any reference numbers):

Our study followed ethical guidelines proposed for the Spanish Royal Decree 1205/2005 on the protection of animals used in experiments and scientific research. The study was carried out in accordance with permits from the Spanish Bird Ringing Centre (Permit Number: 530115), and the regional government of Castilla y Léon (Expte: EP/CyL/298/2016).

The following information was supplied relating to field study approvals (i.e., approving body and any reference numbers):

Regional government of Castilla y Léon, Spain approved the study (Expte: EP/CyL/298/2016).

The following information was supplied regarding data availability:

Raw data, including the genotypes of each individual and matrices of geographic distances between nests, are available in Supplemental Files.

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
