# Peer review of "Genetic signatures of population bottlenecks, relatedness, and inbreeding highlight recent and novel conservation concerns in the Egyptian vulture"

_PeerJ, doi:10.7717/peerj.11139_

## Round 0.1 · original submission · Major Revisions

Dear authors,

Our reviewers suggest a thorough revision of your manuscript. Follow their advice as I will send your revision to them in the next round again.

Greetings
Michael Wink
Academic Editor

Reviewer 1 ·

Basic reporting

The submitted manuscript presents fresh data from a long-term monitoring study of a Spanish population of Egyptian vultures over a period of thirteen years. A comprehensive population genetic analysis is complemented with a screening of several common avian pathogens. Because the largest part of the species‘ European breeding range is on the Iberian Peninsula, these data from actually declining Spanish populations are of a certain importance for conservation management.
The manuscript is well written and structured. However, the discussion is a single long paragraph that could benefit from some substructuring by some subheadings.
To my impression reference to previous work could be a little more substantial. In l. 83 it is mentioned that „information on genetic diversity of populations and species is still limited“. Though the papers by Agudo and collaegues on the Egyptian vulture are mentioned later in the manuscript, there are many more population genetic studies on other vulture species, such as Davidovic et al. (2020) Sci Rep 10: 20394; Poulakakis et al. 2008 Biol J Linn Soc 95, 859–872; Çakmak et al. 2018: Ibis 161:793-805; Godoy et al. 2004: Mol Ecol 13: 371-390. This should be taken into account.

Experimental design

My major concern relates to the use of microsatellites for kinship analyses. Microsatellites have been in use for population genetic analyses for a long time, however, recently there is a growing general criticism on the use microsatellite data (single tandem repeats, STR) for specific research questions for a number of reasons. Indeed, I believe that micosatellites are useful tools for population genetic analyses and in my research group we have used this approach for a number of hybrid zone studies – which has been severly criticized during review processes with the argument that STR data would not be appropriate for the study of hybridization patterns between species – which I think is a false argument. In that study we did not even intend at identifying hybrid classes (F1, F2, backcrosses etc.) with our microsatellite data set, and I would admit that this would have been much more problematic. For example, when we compared our microsatellite data set from one hybrid zone with SNP data set for the same sampling we realized that indeed hybrid classes could be more reliably discriminated against each other using genome-wide SNPs. So I generally believe that microsatellites are useful for analyses of particular population genetic questions whereas they might have limitations with respect to others (e.g. indentification of hybrid classes).
There is a similar criticism on the application of microsatellites for kinship analyses and I feel that this problem needs to be adressed in a revision. There is evidence from several kinship studies that micsats performed less reliably than SNP data sets particularly for groups without parental information, for example: Lemopoulos et al. (2019: Ecol Evol 9:2106–2120.: SNPs „outperformed microsatellites for applications that required individual‐level genotype information, such as quantifying relatedness“); Hauser et al. (2011: Mol Ecol Res 11 (Suppl. 1):150-61); Thrasher et al. (2018: Mol Ecol Res 18: 953-965); Weinmann et al. (2014: Mol Ecol Res 15:502-511); Flanagan & Jones (2019: Mol Ecol 28:544-567). Therefore, the choice of methods needs any kind of justification – in the current version the method is presented straightforward without any critical comment.

Likewise, the QG estimator has been greatly criticized and several alternatives have been suggested (see discussion in Konovalov & Heg (2008: Mol Ecol Notes 8: 256-263). Moreover, this estimator has been shown to become biased when calculation of allele frequencies is based on small sample-sizes (Wang 2017: Herdedity 119:302-313). In the submitted study the pedigree analyses are based on ten individuals (over time, four breeding seasons; 14 individuals in 2018; Fig. 1) … which indeed must be considered small sample sizes. This general criticism should be taken into consideration and a revised methods paragraph should be complemented with a strong justification for the choice of this particular estimator for genetic relatedness.

Demographic events: I agree that bottleneck analysis as done here may be appropriate for local samplings to estimate recent historical events over past decades, however, I doubt whether it makes sense trying to estimate past bottlenecks „over centuries and millennia“ with one local population sampled over a period of 13 years. For this aim, a larger range-wide sampling of a metapopulation would be more appropriate. Therefore, I would indeed recommend to keep the short-term analysis with BOTTLENECK but to discard the M-ratio analysis.

Sample size:
l. 179: „Overall, our study includes the sampling of 42 Egyptian vulture individuals.“ In the text are mentioned: 19 from 2006, 14 from 2018 (l. 173) and further 10 to study relatedness among nestlings (l. 185: eight nestlings and two adults as shown in Fig. 3) – so these sum up to 43 … something must be wrong here.

Validity of the findings

To my impression the validity of the findings depends on a strong justification for the chosen methods to estimate genetic relatedness and an appropriate discussion of potential pitfalls when microsatellites are used for this kind of analyses (see my comments above).

Additional comments

Some minor comments refer to rather formal and editorial issues like typos, correctness of citations etc.

l. 89: „a species‘ range“
l. 99: Once in the main text, the scientific name of the study organism should be mentioned; probably here at first mention (apart from the abstract).
l. 149-151: Have these data on dispersal distance been published? It seems that this information is part of this study’s results (Fig. 1B).
l. 43: „Frankham 2002“ is not in the reference list – is it „Frankham et al. 2002“ or „Frankham 2010“? Please, check the references carefully throughout.

·

Basic reporting

I like the way the article is written. In general, I would, however, shorten the introduction on the theory behind the analyses (as the paper focusses on the conservation and genetics of the organism rather than the theory) and move the "Study species and study area" section into the introduction, to have the introduction set the stage for the analyses, results and discussions. The manuscript is written in good english over all, but I have some smaller correction to the english, which I will point out in detail below.

Experimental design

The study is well designed. However, what is not clear to me is the number of individuals included in the "screening of infectious agents". Why only include two individuals if you later state in the paper that you typed many individuals (which you cite as unpublished data). Given that effectively only two individuals were typed the assumptions in the Discussion are too strong. I really enjoyed the inclusion of that test in the study, and would - if possible - suggest to include the data from the other individuals in this manuscript. If that is not possible, I would recommend to remove the ascertainment that there might be an "interaction between inbreeding and malformation due to BFDV" as this is based on too few samples.

Validity of the findings

The findings are important for the conservation of the Egyptian vulture. I enjoyed the comparison between the two sampled years, which is very valuable for conservation management, and in general the sample sizes are adequate (except for the screening for infectious agents). I am not an expert on demographic inference using microsatellite data, but the methods sounded very reasonable and thought through to me.

Additional comments

I really enjoyed reading the manuscript and think that with some adjustments to make the introduction more focused on the biology rather than the theory, this will be a very important contribution to the conservation genetic literature.

Please, find specific minor comments below:

Title and Line 31: Replace "alert on" with "highlight".
Introduction: Please, reduce the discussion of the theory and include information on the species and area here rather than the M&Ms.
Line 105: Replace "to deepen on the extend of population inbreeding" with "to further investigate inbreeding"
Line 122-123: Add reference.
Line 133-134: Add reference.
Line 145-149: Add reference.
Line 185: Do you mean you sampled 10 individual in each of this years? Please, make that more clear.
Line 204: This might be a conversion issue, but the "10 l" and the "5 l" are missing the micro before the l.
Line 233: Change to "9,999"
Line 260: Change to "9,999"
Line 270: Change to "... based on different assumptions. BOTTLENECK is ..."
Line 302: Change to "The inbred"
Line 334: Wouldn't it have been more logical to remove it?
Line 344:Change "nestling" to "nestlings"
Line 349: Change to "when the offspring"
Line 360: Change to "correctly assigned"
Line 365 and 372: Change to "nestlings"
Line 394: Change to "were 100"
Line 423: Please, change to "failed to reproduce in 2017"
Discussion: The discussion would highly benefit from headers for the different sections.
Line 446-447: Add reference.
Line 454: Please, change to "two breeding seasons twelve years apart"
Line 457: Remove "by", it is either "by" or "using" not both.
Line 476: Same, change to "using"
Lien 502: I would add the years to "all and most" - otherwise it reads a bit odd
Line 506, 507 and 508: Change to "related to"
Line 550: Change to "could have been born".
Line 569: Why would the malformations have changed in captivity if they are birth defects? Did the individual die after two years in captivity?
Line 576: "positive for"
Line 580: This is too strong a statement based on a sample size of 1 in each group. Please, rephrase.
Line 582-584: Please, add some data on the BFDV-positive nestlings or remove the sentence. I think that data would really improve the study as this is an important finding. In general, there is way too few data and not enough actual testing to infer an interaction between inbreeding and malformation and BFDV.
Line 585: Change to "includes" - as this is a general statement.
Line 595-597: I would remove the sentence. It doesn't really fit the topic of this section.
Line 602: Change "alert on" to "highlight"

I am really looking forward to see this manuscript published and hope that my suggestions will help to make it even more enjoyable to read. I am thus recommending the paper for "minor revisions".

Good luck with this and future conservation projects!
Stefan Prost, PhD

---

## Round 0.2 · accepted · Accept

Dear authors,

Many thanks for your revision which followed the recommendations of the reviewers. The critical points have been well revised. Your ms is now acceptable and ready for the next steps.

Greetings
Michael Wink
Academic Editor

Reviewer 1 ·

Basic reporting

The authors have submitted a thorough and comprehensive revision of their manuscript under consideration of all recommendations of the reviewers. Further literature was included in the introduction as recommended. Likewise, the reference to previous work was completed. I particularly appreciate the new figures added to the study. This is a great improvement as compared to the first version and provides a comprehensive documentation and presentation of the analyses and results. Well done!

Experimental design

The research topic and basic research questions were already well defined in the first version. The additional discussion on the explanatory power of microsatellite data (despite all limitations) as compared to other sets of molecular markers and further information on methodical details of genetic relatedness and inbreeding measures add further support to the results.

Validity of the findings

The revised manuscript is complemented with an extensive set of supplementary data that substantiate the results. I have no objections against the revised version.

Additional comments

I would like to congratulate the authors to a fine study and a greatly improved manuscript.

·

Basic reporting

I have read the replies to the comments and the manuscript and I am very happy with the changes.

The only comment I have, is to remove the sentence "This suggests a potential interaction between inbreeding and malformation depending on infection by BFDV, although other non-surveyed pathogens or pollutants could be involved in disease causation, which requires further research." - as a sample number of N=1 is too small to suggest any interaction.

I am looking forward to see the paper published and to follow any upcoming research on this Egyptian vulture population.

All the best,
Stefan Prost, PhD

Experimental design

All good.

Validity of the findings

All good.